# A Polyherbal Mixture with Nutraceutical Properties for Ruminants: A Meta-Analysis and Review of BioCholine Powder

**DOI:** 10.3390/ani14050667

**Published:** 2024-02-20

**Authors:** Germán David Mendoza-Martínez, José Felipe Orzuna-Orzuna, José Alejandro Roque-Jiménez, Adrián Gloria-Trujillo, José Antonio Martínez-García, Nallely Sánchez-López, Pedro Abel Hernández-García, Héctor Aaron Lee-Rangel

**Affiliations:** 1Departamento de Producción Agrícola y Animal, Universidad Autónoma Metropolitana—Xochimilco, Mexico City 04960, Mexico; gmendoza@correo.xoc.uam.mx (G.D.M.-M.); jroque@correo.xoc.uam.mx (J.A.R.-J.); agloria@correo.xoc.uam.mx (A.G.-T.); jamgar@correo.xoc.uam.mx (J.A.M.-G.); md.nallelysanchez@gmail.com (N.S.-L.); 2Departamento de Zootecnia, Universidad Autónoma Chapingo, Chapingo 56230, Mexico; jforzuna@gmail.com; 3Instituto de Ciencias Agrícolas, Universidad Autónoma de Baja California, Ejido Nuevo León, Mexicali 21705, Mexico; 4Centro Universitario Amecameca, Universidad Autónoma del Estado de México, Amecameca 56900, Mexico; 5Facultad de Agronomía y Veterinaria, Centro de Biociencias, Instituto de Investigaciones en Zonas Desérticas, Universidad Autónoma de San Luis Potosí, S.L.P., Soledad de Graciano Sánchez 78000, Mexico; hector.lee@uaslp.mx

**Keywords:** nutraceuticals, polyherbals, plant feed additives, choline, meta-analyses

## Abstract

**Simple Summary:**

BioCholine Powder can improve domestic ruminants’ (sheep, goats, calves, and cows) productive performance. The objective of this study was to review published data and evaluate the effects of dietary polyherbal supplementation in lambs, ewes, dairy goats, and cows on productive performance and blood serum metabolites through a meta-analysis and by a comparison of the estimated net energy for maintenance (NEm) and gain (NEg) in lambs, which was expressed as a percentage of change in the polyherbal mixture regarding their control using a Chi-squared test. BioCholine supplementation improved small ruminants’ daily gain and milk production and modified some blood metabolites. The results confirm that including BioCholine Powder in ruminant diets shows nutraceutical effects that outweigh those of phosphatidylcholine bypass and that benefits in growth, milk production, and health can therefore be expected with its dietary inclusion.

**Abstract:**

BioCholine Powder is a polyherbal feed additive composed of *Achyrantes aspera*, *Trachyspermum ammi*, *Azadirachta indica*, and *Citrullus colocynthis*. The objective of this study was to analyze published results that support the hypothesis that the polyherbal product BioCholine Powder has rumen bypass choline metabolites through a meta-analysis and effect size analysis (ES). Using Scopus, Web of Science, ScienceDirect, PubMed, and university dissertation databases, a systematic search was conducted for experiments published in scientific documents that evaluated the effects of BioCholine supplementation on the variables of interest. The analyzed data were extracted from twenty-one publications (fifteen scientific articles, three abstracts, and three graduate dissertations available in institutional libraries). The studies included lamb growing–finishing, lactating ewes and goats, calves, and dairy cows. The effects of BioCholine were analyzed using random effects statistical models to compare the weighted mean difference (WMD) between BioCholine-supplemented ruminants and controls (no BioCholine). Heterogeneity was explored, and three subgroup analyses were performed for doses [(4 (or 5 g/d), 8 (10 g/d)], supplementation in gestating and lactating ewes (pre- and postpartum supplementation), and blood metabolites by species and physiological state (lactating goats, calves, lambs, ewes). Supplementation with BioCholine in sheep increased the average daily lamb gain (*p* < 0.05), final body weight (*p* < 0.01), and daily milk yield (*p* < 0.05) without effects on intake or feed conversion. Milk yield was improved in small ruminants with BioCholine prepartum supplementation (*p* < 0.10). BioCholine supplementation decreased blood urea (*p* < 0.01) and increased levels of the liver enzymes alanine transaminase (ALT; *p* < 0.10) and albumin (*p* < 0.001). BioCholine doses over 8 g/d increased blood glucose, albumin (*p* < 0.10), cholesterol, total protein, and globulin (*p* < 0.05). The ES values of BioCholine in retained energy over the control in growing lambs were +7.15% NEm (*p* < 0.10) and +9.25% NEg (*p* < 0.10). In conclusion, adding BioCholine Powder to domestic ruminants’ diets improves productive performance, blood metabolite indicators of protein metabolism, and liver health, showing its nutraceutical properties where phosphatidylcholine prevails as an alternative that can meet the choline requirements in ruminants.

## 1. Introduction

Choline is involved in three main metabolic pathways, the donation of methyl groups, the synthesis of acetylcholine, and phosphatidylcholine (Ptdcho) [1], and can be obtained from the diet or synthesized de novo [2], which has made it difficult to determine the precise choline requirements in ruminants [3,4,5]; however, RPC (rumen-protected choline) experiments have shown that choline is a limiting nutrient for domestic ruminants [6,7,8,9], particularly where genetic potential has resulted in an increase in the metabolic nutrient demands. RPC is a commercial product developed to deliver choline to the small intestine for absorption by protecting choline chloride from ruminal degradation, and it is commonly supplemented in dairy farms.

The evidence compiled in three meta-analyses shows that supplementation with (RPC) increases milk production in dairy cattle [6,7,8]. RPC is a commercial product developed to deliver choline to the small intestine for absorption by protecting choline chloride from ruminal degradation, and it is commonly supplemented in dairy farms. In addition, its inclusion in the transition diet and during early lactation might reduce ketosis and mastitis problems [9], thereby improving the antioxidant condition of cows [10]. However, in the nutritional requirements of ruminants [3,4,5], a specific recommendation has not been proposed due to endogenous synthesis [2] and the variability of the products used to provide bypass choline [11,12], which complicates the estimation of requirements. Nevertheless, there are reports of positive responses to RPC supplementation in dairy goats [13], beef cattle [14], and feedlot lambs [14,15].

The possibility that choline might be contained in natural sources that contribute compounds resistant to rumen degradation is very low because it is recognized that choline from feed (mainly in the form of phosphatidylcholine) is extensively degraded in rumen [16,17]. Nevertheless, two abstracts from the American Society of Animal Science Meeting [15,18] reported information about a polyherbal mixture (provider certifies the presence of choline conjugates, mainly Ptdcho), where the results showed positive results equivalent to those of a commercial RPC product. Since these reports, experiments have been published in which the secondary metabolites of the polyherbal mixture have been characterized [19,20], and experimental evidence has shown that both needs can be covered. The physiological requirements of choline in ruminants can be met with the Ptdcho present in polyherbals, as in non-ruminants [21,22], and it is important to consider that feeds can provide free choline and different choline metabolites (choline-contributing compounds), such as betaine, glycerophosphocholine, phosphocholine (PCho), Ptdcho, and sphingomyelin [23], whose rumen degradation has not been fully characterized but may have different bioavailabilities, as observed in rodent models [24].

The metabolic requirements for choline (for betaine or acetylcholine) can be derived from phosphatidylcholine [25] via two pathways described by Fagone and Jackowski [26]. The first is when the choline group is replaced with serine by the reaction catalyzed by phosphatidylserine synthase 1, producing phosphatidylserine and choline; the second is by the hydrolysis of Ptdcho into choline and phosphatidic acid by phospholipase D. It can be expected that when supplying different amounts of Ptdcho, the relative balance between the use of choline as a methyl donor (via betaine) and the acetylcholine precursor (via choline) or phospholipid precursor (via PCho and Ptdcho) can be modified, as observed in rodent models [24], limiting the need for free choline and the energy expenditure associated with its absorption and those required for the synthesis of Ptdcho from choline.

Initial ruminant studies have shown that Ptdcho in pastures is rapidly degraded in the rumen in vitro and in vivo by rumen bacteria [16]. Therefore, there have been few evaluations of dietary phospholipids in ruminants. However, a comparison of soy lecithin and RPC that lacked a strict control group [27] showed the potential of using choline-contributing compounds. The experiments carried out with BioCholine polyherbal mixture showed that Ptdcho resists ruminal degradation, which was confirmed with blood levels of Ptdcho in lambs and by better productive parameters [28,29], methylation [20], and other evidence described in this document.

There is no conventional information on the in situ rumen degradation of choline-contributing compounds present in the BioCholine polyherbal mixture, and this methodology cannot be used to determine the amount of Ptdcho that escapes rumen degradation by microbial synthesis of phospholipids. Therefore, the physiological evidence in blood or gene expression, as well as the productive parameters of ruminants fed with BioCholine Powder, are variables that can demonstrate its bypass metabolites and its nutraceutical properties [20,30,31,32]. Therefore, the objective of this document was to analyze published results that support the hypothesis that the polyherbal product BioCholine Powder has nutraceutical properties and phosphatidylcholine as an alternative to meet choline requirements through a meta-analysis and effect size analysis. 

## 2. Materials and Methods

### 2.1. Data Analyses

The experiments’ identification, selection, and inclusion were conducted following the PRISMA methodology [33] as described by Orzuna-Orzuna et al. [34], as shown in Figure 1. Scientific articles were searched in Scopus, Web of Science, Science Direct, and PubMed databases with BioCholine fed to ruminants to integrate information on the product of the BioCholine Powder. Dissertations and abstracts from scientific meetings were also included. Documents that did not measure the variables of interest were excluded. In order to be considered, studies had to meet several inclusion criteria, as described by other authors [33,34]. For growing ruminants, dry matter intake (DMI), average daily gain (ADG), feed conversion ratio (FCR), and initial and final body weight (BW) were considered; for lactating and gestating ruminants, milk production (MP), body weight (BW), lactation, DMI, FCR, and chemical composition of experimental rations were considered. We also included studies that carried out quantification or possible determination of daily BioCholine dietary intake, including least squares means of the control and experimental groups with the standard error or standard deviation and the number of experimental units.

The analyzed data were extracted from twenty-one publications (fifteen scientific articles, three abstracts, three graduate dissertations available in institutional libraries) and one unpublished dataset from our research team (Table 1). The original data on weight changes of ewes and their offspring and milk yield from the Roque-Jiménez et al. [20] experiment were also re-analyzed to include results in the meta-analysis and the effect size meta-analysis. The PRISMA methodology [33] was followed, as described by Orzuna-Orzuna et al. [34], for the identification, selection, and reference inclusion process (Figure 1), using the following keywords: BioCholine, blood metabolites, calves, choline, dairy cow, digestibility, energy metabolites, feed additives, feed plant additive, gene expression, goats, growing sheep, herbal, herbal choline, Holstein calves, lambs, milk quality, polyherbal, pregnancy, rumen fermentation, rumen-protected choline, and sheep. The selection process resulted in studies published between July 2015 and October 2022.

### 2.2. Meta-Analysis and Subgroup Analysis

Meta-analysis and subgroup data were analyzed as described by Orzuna-Orzuna et al. [34] using the Metafor package [50] of R statistical software (v.4.1.2, R Core Team, Vienna, Austria). The effects of the dietary inclusion of BioCholine were evaluated through weighted mean differences (WMDs) between diets without BioCholine (control treatments) and diets with the inclusion of BioCholine (experimental treatments). The WMD treatment means were weighted by the inverse of the variance for random effects models proposed by DerSimonian and Laird [51]. Heterogeneity was measured using the I^2^ statistic and Chi-squared (Q) test [52]. A significance level of *p* ≤ 0.10 was used in the Q test [53]. The I^2^ values were interpreted as follows. (1) I^2^ < 25% indicates low heterogeneity, (2) I^2^ between 25 and 50% indicates moderate heterogeneity, and (3) I^2^ > 50% indicates high heterogeneity [54].

Sources of heterogeneity for all response variables were assessed by subgroup analysis [52]. Covariates were evaluated using subgroup analysis by dividing the covariates as follows: lamb daily intake of BioCholine ≤5 and >5 g; lactating small ruminants’ daily intake of 4 (or 5) g/d, 8 (or 10) g/d, and supplementation period (pre- and postpartum), blood metabolites by BioCholine daily intake ≤4, 5–8, and >8 g, and species (goats, calves, lambs, and ewes).

### 2.3. Size Effect Comparison 

Since not all variables could be included in the meta-analysis, the effect size (ES) of BioCholine [34] on changes in estimated net energy was compared, expressing results as the percentage change based on weighted mean SE, which was compared with the Chi-squared test using the sum of the n of treatment and its controls [55] using MedCalc Version 22.003 statistical software.

### 2.4. Net Energy Estimations

For the experiments of growing lambs with BioCholine and RPC, the NEm and NEg from each treatment and its controls were estimated from the reported data of body weight (BW), average daily gain (ADG), and dry matter intake (DMI), as described in lambs [56,57]. Calculations were based on shrunken body weight (SBW), estimated at 96% [58]. The maintenance energy requirement (Mcal/d) was estimated according to the NRC [59] formula ME = 0.056 BW^0.75^ and gain energy (GE; Mcal/d) with the formula GE = 0.276 × ADG × SBW^0.75^. The actual net maintenance energy (NEm) and net gain energy (NEg) were calculated from maintenance energy (ME), gain energy (GE), and dry matter intake (DMI) using the following equations published by Zinn et al. [60]: a = −0.41 × ME; b = 0.877 × ME + 0.41 CMS + ER; and c = −0.877 × DMI, substituting the values in the formula Nem = (−b − √ (b2 − 4ac))/2c to obtain the actual NEm (Mcal/kg) and in the equation Neg = (0.877 × NEm) − 0.410 to obtain the actual NEg (Mcal/kg). The percentage difference between net energy changes in BioCholine values was compared with a Chi-squared test [55], with WMD weighted according to the n values of the treatments and controls.

### 2.5. BioCholine Review 

A review of the composition of secondary metabolites present in BioCholine was made, as well as experiments with graded levels and effects of the polyherbal on ruminal fermentation, fertility, animal health, gene expression, and methylation, to understand the mechanisms of action and the results observed with the polyherbal in domestic ruminants. These variables could not be included in the meta-analysis.

## 3. Results and Discussion

### 3.1. Polyherbal (BioCholine) Characteristics

BioCholine Powder Polyherbal is a polyherbal mix made from Indian species, such as *Achyrantes aspera*, *Trachyspermum ammi*, *Azadirachta indica*, and *Citrullus colocynthis*, and Andrographis. It is a standardized feed plant additive produced by Indian Herbs Specialties Pvt Ltd. (Uttar Pradesh, India), with certifications including ISO 9001 [61], GMP, and GMP Plus. It is marketed in some European countries and almost all Latin American countries by Nuproxa Switzerland Ltd., (Etoy, Switzerland) [19]. 

It differs from synthetic protected choline products in that it provides Ptdcho and not choline chloride; besides being phytobiotic, it offers secondary metabolites with different nutraceutical properties [21]. It is not an herb spice extract, so it preserves other active molecules (some with greater predominance) and nutrients (or precursors) that explain the biological effects observed in supplemented animals [62].

Some secondary metabolites found in BioCholine have antimicrobial effects [19,20,63] that may give it resistance to ruminal degradation of choline conjugates. Mendoza et al. [19] analyzed the volatile compounds of the mixture by flash gas chromatography electronic nose. These authors reported 15 relevant compounds, including aromas, alcohols, phenolics, and aldehydes, with compounds that could reduce microbial activity, such as aldehyde β-pins, Trans-2-Undecenal, and 1-propanol. Roque-Jiménez et al. [20] characterized BioCholine Powder using gas chromatography coupled with mass spectrophotometry and reported 19 organic compounds. Some of these compounds belong to the methyl groups hexadecenoic acid methyl ester (C16:0) and octadecenoic acid methyl ester (C18:1 cis 9, 12; C18:1 cis 9; C18:1 cis 8). Orzuna-Orzuna et al. (unpublished data from a doctoral dissertation at Universidad Autónoma Chapingo Postgraduate in Animal Production; Appendix A) used a gas chromatography–mass selective detector and confirmed the predominance of C18:1 cis 9, 12 (58.94%) and the presence of C16:0 (16.24%) and some C18:1 cis 9 (0.98%). These have been associated with fatty acid methyl esters with DNA methyl groups; another important compound is thymol, which can prevent acetylcholine reduction and increase recycling. Some components, such as thymol [64,65,66], Phenol, 4-methoxy-2,3,6-trimethyl [67], and 9.17-Octadecadienal (Z) [68], have antimicrobial activity.

The supplier of the polyherbal reports content of choline conjugates between 1.76 and 1.8%, a higher value than that reported by Leal et al. [43] of 0.978% analyzed with High-performance Thin Layer Chromatography, with total phospholipids of 1.68%. Dazuk et al. [63], using high-performance thin-layer chromatography, reported phosphatidylethanolamine, phosphatidylinositol, phosphatidylcholine, lysophosphatidylcholine, and 16.1% PTCho natural choline conjugates. The polyherbal has other metabolites that can affect the rumen microbiota and the metabolism of ruminants. Roque-Jiménez et al. [20] detected the presence of compounds containing methyl groups, such as hexadecenoic acid methyl ester (C16:0) and octadecenoic acid methyl ester (C18:1 cis-9; C18:1 cis-8), related to the methylation status of DNA. BioCholine also contains tannins and flavonoids [63].

### 3.2. Gene Expression

Nutrigenomic studies show the effects of BioCholine expressed at the cellular level, allowing us to explain changes in productive variables, health, and antioxidant status. In the calf experiment [41], changes in leukocyte expression were evaluated by microarrays from blood samples collected on day 60 of the experiment, when the calves consumed 4.95 ± 0.686 kg of starter concentrate per day and the milk had been suspended for 25 days, so the rumen was adapted to eating solids, and the response could be interpreted as calves with a developed rumen (BW 93.21 ± 9.53). Microarray results showed that 1093 genes were upregulated and 1349 were downregulated with BioCholine, with marked changes in 13 genes related to lipid metabolism, 9 genes related to carbohydrate metabolism, and 9 genes related to oxidation–reduction. 

The review of expression change in specific genes allows us to explain some beneficial effects of BioCholine; among the genes that were overexpressed were *PPARα* (peroxisome proliferator-activated receptor Alpha, +2.82), *Acoxl* (acyl-coenzyme A oxidase-like, +4.0), *Gck* (glucokinase, +2.06), *TGFβ-1* (transforming growth factor, beta 1, +3.40), and *Defa 14* (defensin, alpha 14, +3.0), and among those that were under-regulated were *G6pc3* (Glucose 6 phosphatase catalytic 3, −3.80), *G6pc3* (glucose 6-phosphatase catalytic 3, −3.76), and *Ugdh* (UDP-glucose dehydrogenase −2.01) [41].

The *PPARα* gene has implications for the oxidation of fatty acids in ketogenesis, and its effects result in lower blood triglyceride levels [30]. In contrast, the genes related to glucose metabolism are involved in the homeostatic regulation of blood glucose levels [69]. The *Acoxl* gene is involved in peroxisomal fatty acid beta-oxidation [70] in the metabolism of very long-chain fatty acids [71]. Genes of the *TGF-β* family have essential roles in tissue development, cell activity, and bone metabolism, as well as in the control of the immunological response, healing process, and inflammatory response [72]. The *Defa 14* gene is an antimicrobial peptide found in the skin, intestinal mucosal, and respiratory tract and has been related to antimicrobial and antiviral properties [73]. Ontological analysis and specific genes make it possible to explain the improvements in health and immune response observed in the experiments in which these indicators were evaluated [29,41,47].

Other nutrigenomic evidence derived from a growth-fattening experiment with Pelibuey lambs showed changes detected by microarrays in liver samples, which are indirect indicators of the presence of bypass metabolites (Orzuna-Orzuna et al., unpublished data from the doctoral dissertation). In these lambs, BioCholine differentially modified the expression of 2312 genes, of which 1135 were downregulated and 1177 were upregulated by at least 1.5 times compared to the control lambs. The ontological analysis showed that BioCholine stimulated several pathways, including folate biosynthesis, nucleotide excision repair, oxidative phosphorylation, endocrine resistance, platelet activation, thermogenesis, and chemokine signaling pathway, and reduced others, such as the TGF-beta signaling pathway and fluid shear stress. These pathways provide insights into cellular processes that affect metabolism and health influenced by the polyherbal. 

The expression of specific genes provides additional information that helps explain the benefits of the polyherbal. The genes oxidative phosphorylation NADH–ubiquinone oxidoreductase subunit and NADH–ubiquinone oxidoreductase subunit A1 are overexpressed 2.66 and 2.47 times, respectively, and these genes are involved in mitochondrial integrity and the cell’s antioxidant defense systems against cellular reactive oxygen species (ROS) [74]. The polyherbal also promotes higher expression of the SOS Ras/Rho guanine nucleotide exchange factor 2 (+2.5) gene, which is involved in cell signaling conditions [31]. Another gene stimulated +1.99 times is glutathione S-transferase, mu 2, which has hepatoprotective effects [75]. Therefore, the antioxidant and detoxifying capacities were improved with BioCholine. 

The *leptin* gene is overexpressed 1.59 times and is an essential hormone in energy metabolism and intake [76], with important physiological effects. Other genes are under-expressed, such as transforming activin receptor IIA growth factor (−2.78 times), beta receptor II (−2.5 times), latent transforming growth factor beta binding protein 1 (−2.33 times), and transforming growth factor, beta 1 (−2.09), which participate in the TGF-beta signaling pathway regulated through feedback mechanisms that control the magnitude of its signals [77].

Microarrays allow for a better understanding of the genome and transcriptome, as shown by evaluating all genes in the phenomenon studied [78]. The overall results in pathways allow us to explain the improvements in performance parameters in sheep [28,35,37,43,44], antioxidant status [36], and immune response and cattle health [41,47].

Another study carried out on sheep by Roque-Jiménez et al. [20] presented evidence of the effects of BioCholine in fetal and integral nutritional programming. This study supplemented ewes during different thirds of gestation and all gestation using 4 g BioCholine per day of each third and throughout gestation and evaluated the epigenetic modulation of 5-hmC in whole blood from the supplemented ewes. The 5-hmC DNA marks are intermediate in DNA demethylation and are stable epigenetic marks. Roque-Jiménez et al. [20] indicated that supplementation throughout gestation led to a greater percentage of 5-hmC during pregnancy; higher percentages of 5-hmC DNA in offspring born from supplemented ewes were also observed, and those significant changes in the percentages of 5-hmC in whole blood were attributable to BioCholine bio-active substances (bypass metabolites). 

The higher methylation in sheep supplemented with BioCholine can be explained by the contribution of labile-free methyl groups for DNA methylation and the presumable saving of choline by thymol, which promotes acetylcholine recycling by inhibiting acetylcholinesterase activity (Figure 2) [79]. Other studies [80,81] provide evidence that a few compounds reported in BioCholine serve as methyl donors and modify the methylation status of DNA, e.g., hexadecenoic acid methyl ester (C16:0), octadecenoic acid methyl ester (C18:1 cis-9; C18:1 cis-8), and thymol. The effects on ewes and their offspring indirectly indicate that some BioCholine metabolites resist rumen degradation. These metabolites are absorbed in the lower tract, transported by blood, and modulate the epigenetics of the ewes. Because the polyherbal has many metabolites, we hypothesize that the response is due to Ptdcho and methylated compounds (Figure 2). However, further research must demonstrate how metabolites are transported and stored in different tissues.

### 3.3. Rumen Fermentation

One experiment with ewes evaluated the effects of BioCholine on rumen fermentation [43], where no changes in rumen pH, acetate, and total VFA were reported. Compared to the control, supplementation with 4 and 8 g/d of BioCholine reduced the ruminal propionate concentration by 24.0 and 17.3% in lambs sampled on days 15 and 45 of the experimental period. Butyrate was reduced by 30.5% in the samples from the 8 g/d group. Nunes et al. [46] also evaluated ruminal fermentation in dairy cows fed increasing levels of BioCholine (0, 7, 14, and 21 g/d) and did not observe differences in ruminal VFA concentration and molar proportions, NH_3_-N, nitrogen balance, and digestibility (DM, crude protein, NDF, and ether extract); however, ruminal pH and OM digestion showed a quadratic reduction (*p* < 0.10), with the lowest values at 7 g/d, which coincided with the quadratic response in milk production (*p* < 0.05).

The statistical differences observed by Leal et al. [43] in butyrate and propionate have little impact on fermentation efficiency since a minimal CO_2_ reduction (1.45% of the control) and a marginal methane increase (2.67%) are expected when including a polyherbal. The indicator of microbial activity (methylene blue test) showed a significant boost with herbal supplementation (24.1 and 50.2% for doses 4 and 8 g/d) [43]. The in vitro gas production of BioCholine samples shows that herbal choline has three ruminal degradable fractions, one with fast ruminal degradation (31.8%), another with a medium rate of degradation (40.7%), and one with a slow rate of degradation (27.5%), indicating that an important fraction of the BioCholine Powder can reach the lower tract [48]. Further studies of ruminal fermentation and changes in rumen protozoa and bacteria with metagenomics are required to determine other changes due to the secondary metabolites of BioCholine.

### 3.4. Meta-Analyses 

The meta-analysis included studies conducted in two countries, México (66.7%) and Brazil (33.3%), and experiments were conducted on commercial farms (37.5%) and experimental facilities (62.5%). The experimental doses of BioCholine in small ruminants ranged from 1.5 to 15 g/d and 7 to 40 g/d in dairy cattle. The experimental periods ranged from 42 to 59 days for growing assays, 20 to 183 days in lactating small ruminants, and 60 to 1095 days in dairy cows. 

Dietary inclusion of BioCholine increased (*p* < 0.01) the average daily gain (ADG), final BW, and milk production (Table 2). However, dietary supplementation of BioCholine did not affect (*p* > 0.05) the dry matter intake (DMI) or feed conversion ratio. The I2 values for DMI, milk yield, feed conversion, and final BW indicated high heterogeneity between studies [54,82], whereas ADG had the lowest heterogeneity.

Subgroup analysis of doses in feedlot lambs’ diets (Table 3) showed that regardless of the amount of BioCholine, ADG was increased (*p* < 0.05), which was also expressed in the heaviest final BW (*p* < 0.10) due to a higher obtention of dietary energy than the controls.

The milk production of small ruminants tended to increase (*p* = 0.11) with BioCholine doses between 4 and 5 g/d (Table 4), and prepartum supplementation significantly improved milk production (*p* < 0.10).

Ruminant blood tests have been reported in 12 experiments [28,35,36,37,38,41,42,43,44,45,48,49]; unfortunately, the blood concentration of Ptdcho was evaluated (evidence of exceeding the metabolites of Ptdcho) only in one experiment [28], whereby increasing the dose of BioCholine caused a linear increase in blood Ptdcho. The other metabolites shown are indicators of effects in the animal due to the polyherbal.

The meta-analysis of blood metabolites is presented in Table 5. BioCholine reduces the concentration of urea (*p* < 0.05) and increases albumin (*p* < 0.01) and ALT (*p* < 0.10). The urea concentration is related to protein metabolism and other factors related to protein degradability in the rumen [83].

The liver enzymes in the experiments were within the physiological values for sheep, which were healthy. Cross-reactions from some metabolites from BioCholine could be responsible for increasing ALT rather than liver injury [84]. 

BioCholine dose subgroup analysis (Table 6) showed that when consumption was over 8 g/d glucose (*p* < 0.10), cholesterol (*p* < 0.01), plasma proteins (*p* < 0.05), and globulin (*p* < 0.05) increased. Consumptions below 4 g/d reduced the urea concentration (*p* < 0.01).

Ruminant types showed some differential responses in blood metabolites (Table 7). The fattening lambs showed an increase in glucose with BioCholine (*p* < 0.05), while the rest showed a hypoglycemic effect (ewes, *p* < 0.10; goats and calves, *p* < 0.01). The changes observed with glucose metabolism suggest that it will generally have a hypoglycemic effect, but in fattening lambs, energy consumption and the glucose precursor propionate explain the increase [28]. BioCholine contains *Azadirachta indica*, which has shown hypoglycemic effects [85].

Cholesterol was reduced in lactating goats (*p* < 0.01) but increased in lambs (*p* < 0.05; Table 7). Cholesterol increased in four experiments, highlighting the increases in lambs (females and males); in two, a reduction in small lactating ruminants was observed [45,49]. The β-OH-butyrate only increased in two experiments with lactating ewes [42] and decreased in dairy calves [41]. Blood levels could not be related to changes in ruminal butyrate because BioCholine significantly reduced the butyrate concentration [43]. The changes in cholesterol confirm the lipotropic effects of BioCholine and are consistent with the increased expression of *PPARα* and *Acox1* genes [41]. *PPARα* increases fatty acid oxidation in the liver [86]. Blood cholesterol depends on the mobilization and synthesis of fats and is related to phosphatidylcholine, which participates in synthesizing and exporting triglycerides in very low-density lipoproteins [87]. 

The globulin values were not consistent. Some experiments showed increases [36,37,38,43], and others showed reductions [28,35,41,49], but the meta-analysis allowed us to identify that with doses >8 g/d, globulin concentrations increased. The changes in total protein, globulins, and urea reduction suggest that BioCholine affects protein metabolism, presumably in the liver. Blood proteins are an indirect nutritional indicator because they provide amino acids. Primary proteins are produced in the liver [88]. The best status of protein metabolism coincides with the changes in the oncologic pathway of ribosomes, which increased twice in calves that received BioCholine [41].

### 3.5. Comparison of BioCholine and RPC in Small Ruminants

Three experiments comparing BioCholine with RPC Reashure have been reported, one in peripartum and postpartum ewes and two in feedlot lambs. In both experiments, the doses were 4 g/sheep/day of BioCholine vs. 4 g/d of RPC. Crosby et al. [18] used 24 Rambouillet ewes supplemented for 30 days before and after calving with a control group, and milk yield differed (*p* < 0.05) from the control (1.02 kg/d) with both sources: BioCholine (1.57 kg/d) and RPC (1.39 kg/d) (statistically similar). The results showed that the two choline sources improved calving weight and milk production compared to the control, reflected in higher lamb weights at birth and better weight gains at 30 days, showing that BioCholine can replace RPC and obtain similar results [15].

The two experiments comparing BioCholine and RPC in feedlot lambs showed differences, but the substitution effects were not as strong as those of ewes. In the first experiment, 24 Rambouillet lambs (23.4 ± 1.1 kg initial BW) were housed in individual metabolic cages, and treatments were as follows: control (no choline), 4 g/d BioCholine, and 4 g/d RPC in a completely randomized design, used for 42 days [15]. However, no differences were detected in average daily gain (control 222 g/d; BioCholine 250 g/d; and RPC 290 g/d) or feed intake (control 1.06 kg/d; BioCholine 1.07 kg/d; and RPC 1.22 kg/d). However, the final BW improved with the RPC (35.6 kg), followed by BioCholine (33.9 kg) and the control (32.7 kg).

The second experiment was a repetition of the experiment carried out under the same conditions with 24 Rambouillet lambs (23.5 kg ± 3.17 kg initial BW) housed in individual metabolic cages [44] with the same treatments (control group, 4 g/d BioCholine, and 4 g/d RPC) for 42 days. Lambs fed with the polyherbal improved daily gain (320 g/d) and feed intake (1.32 kg/d) compared to the RPC (222 g/d and 1.06 kg/d) and control groups (290 g/d; 1.26 kg/d).

Since dry matter intake has a determinant effect on weight gain, the estimated values of NEm and NEg based on growth and consumption [60] allow us to corroborate that BioCholine was consistent in the two experiments, allowing us to obtain 4.67% more NEm and 1.62% NEg compared to the control group, while the RPC effect size was lower (Table 8).

### 3.6. Evaluations with Graded Levels of BioCholine in Growing Lambs

Several evaluations with graded levels of BioCholine in sheep have been reported: one with lambs for 52 days [28], three with growing ewes of 75 days [37,43], and two with dairy ewes [42,49], while there is only one reported with RPC in growing lambs [89]. Table 9 shows the estimated values of NEm and NEg of the experiments based on the daily consumption of the polyherbal and the statistical comparison of the weighted means with all data (Table 9). BioCholine increased (*p* < 0.10) the NEm by 7.15% and the NEg by 9.25% over the control. However, it must be considered that the animal response to high doses of polyherbal is quadratic, and negative reactions can be observed [28]. Still, the data suggest consistent and safe responses with intakes of 4 to 8 g/d of BioCholine in small ruminants.

### 3.7. Results of Milk Production in Ewes

In six experiments with ewes [18,20,35,36,42,49], increments in milk production with BioCholine were recorded with different doses and days of supplementation in gestation and lactation, as detected in meta-analyses that included data from ewes and goats (Table 4). Only one experiment did not show a response in milk production with doses of 0, 4, and 8 g/d BioCholine [49]. The overall SE effect was an increment of +11.35% over the control group (Chi-squared 9.22, *p* = 0.0024). 

The results from the experiment from Roque-Jiménez et al. [20] were reanalyzed by orthogonal contrasts to compare ewes in the control group to those in the BioCholine group (contrast I), and the supply during the last third of gestation vs. BioCholine supplementation throughout gestation (Contrast II) and supplementation with BioCholine showed benefits throughout gestation treatment, in which, after delivery, benefits were observed in the energy balance with an increase (*p* < 0.01) in milk production of 25.9% with a reduction in weight loss of 25.7%, increasing the weight of their young at birth by 22.4% (Table 10). These results confirm the effects of choline bypass metabolites and methyl groups that were reflected in greater fetal growth similar to that observed with RPC [90], as well as a better energy balance of the sheep supplemented for longer in gestation with a response effect similar to that observed by Tsiplakou et al. [91] when supplementing RPC (5 g/d) with protected methionine and betaine.

### 3.8. Results of Dairy Goats’ Milk Production

The results of two experiments with dairy goats [39,45] of French and Alpine breeds showed surprisingly high responses in milk yield, both conducted at the same experimental campus. Morales et al. [45] orally supplied 0, 4, and 8 g/d, observing a linear response in performance where the 8 g/d doses showed an increase of up to 54%, and in a second experiment where the 8 g/d dose was used for 30 days before delivery and 90 days of lactation, the observed increase (*p* < 0.05) was 215% [39]. The overall SE effect was +71.84% over the control group (Chi-squared 16.906, *p* = 0.0001). 

Milk yield response in dairy goats was greater than in ewes because dairy goats produce more milk than dairy sheep, which has been associated with differences in energy partitioning and differences in insulin and glucose [92]. Also, there have been reported differences in methylated genes between sheep and goats [32], which could be expressed more by the methylated compounds provided by BioCholine. Some sources of RPC have also been evaluated in goats where one of the experiments did not show a response with 4 g/d for 28 days of lactation in Saanen goats [93]. However, in another study with Etawah-bred goats, the RPC showed a quadratic response with the best increase with 4 g/d of +17% over controls supplemented 52 days before calving and 84 days postpartum [94]. D’Ambrosio et al. [13] observed a response of 14.1% with a supply of 4 g/d supplemented 30 days prepartum and 35 days postpartum to Saanen goats. 

### 3.9. Response to BioCholine in Dairy Cattle

Four experiments with lactating dairy cattle with supplemental BioCholine have been reported. One was with cows under grazing conditions (Holstein and Jersey) with a mean initial production of 18 kg/d (n = 81) for 90 days, in which milk production increased linearly (*p* < 0.05) as the dose of polyherbal was increased from 0, 10, to 20 g/d of polyherbal without affecting cows’ BW or milk composition [40]. In another experiment, high-producing Holstein cows supplemented with BioCholine (20 g/d n = 19, 40 g/d n = 17, control n = 20) and other combinations of polyherbals for 90 days since 30 days postpartum showed no response in milk production or composition; however, health indicators were improved, reducing treatment costs (antibiotics, healing, anti-inflammatories, glucogenesis, hormones, intra-mammary treatments, restorative treatments, and vitamins) and being more profitable than the dose of 40 g/d BioCholine. Beneficial residual effects in veterinary expenses and milk production were also detected up to 90 days post-supplementation [47]. 

Milk production was also evaluated in a Latin square experiment with Holstein cows (163 days in milk and 27.6 kg/d average milk yield) treated with 0, 7, 10, and 21 g/d BioCholine, and a quadratic response (*p* < 0.05) was observed in milk production with the best response at 7 g/d and a reduction in fat (square effect *p* < 0.10) without changes in daily intake [46].

There was also a multiannual study in cows (crossbreeding rotational program using Holstein × Montbeliarde × Swedish Red) with an average production of 36 kg/d, where data from 424 control and 442 supplemented cows were analyzed every year for three years in which BioCholine with 0.071% of dry matter of the diet in the entire dairy herd (target dose of 17 g/d for lactating cows) was compared with data from three years without supplement, conducted at a commercial farm, comparing milk production, health status, and replacement data [29]. The supplementation with BioCholine improved fat-corrected milk production by 1.5% (*p* < 0.001) compared to the average value obtained in the previous years (36.36 vs. 35.80 kg/d) without the polyherbal. Other effects on fertility and health will be discussed later.

The combined analysis of the four experiments for milk production showed an increment (*p* < 0.01) of +0.59 in SE (Table 11), and the differences between experiments suggest that the number of replications, days of the experiment, and production level may be a factor affecting the response to BioCholine. Only in two treatments were numerical values observed below those of the supplemented cows [46,47]. There are known differences in the patterns of DNA methylation in mammary gland tissues within Canadian Holstein cows with different milk compositions [95], and DNA methylation and milk yield can likely explain the different responses to the polyherbal summarized in Table 11.

### 3.10. Responses to Health Indicators by BioCholine

Some of the plants in BioCholine have a wide range of antimicrobial and antioxidant activities through secondary metabolites that stimulate the immune system [96]. As described previously, BioCholine contains volatile metabolites, such as 2-Undecenal, 8-p-menthane diamine, 4-vinylguaiacol, β-pinene, p-cresol [19], and some aldehydes with bacteriostatic and bactericidal effects [97,98,99].

The multiannual evaluation in dairy cattle [29] showed a reduction (*p* < 0.0001) in abortions (15.65 to 7.29%), clinical (*p* < 0.005; 12.59 to 6.95%) and subclinical mastitis (*p* < 0.05; 8.65 to 5.22%), and respiratory disorders (*p* < 0.10; 12.42 to 8.56%). The experiment with Lacaune ewes with 0 and 5 g/d BioCholine showed a reduction in somatic cell counts of 37.2% in the milk of supplemented ewes (*p* = 0.07) [35], and the effect was confirmed in another experiment [36] with lactating Lacaune ewes (0, 5, and 10 g/d) in which somatic counts were reduced by 40.42% compared to the control values (samples collected on days 15 and 20 of lactation). 

Immunoglobulins (Anti-Clostridium IgG) increased linearly (+12.58%) (*p* < 0.10) in calves receiving graded levels of BioCholine over the control, and the number of diarrhea events was reduced by 74.71%, pneumonia by 42.29%, and otitis by 49.29%, which resulted in cost reduction in antibiotic doses by 51.03% [41]. In the same experiment, BioCholine supplementation increased gene expression of *Klra20* (Killer cell lectin-like receptor subfamily A member 20, +3.9), *Pdgfra* (platelet-derived growth factor receptor alpha polypeptide, +3.9), *Defa14* (Defensin alpha 14, +3.3), *Pdgfrl* (Platelet-derived growth factor receptor-like, +3.3), *Lck* (Lymphocyte protein tyrosine kinase, +3.2), and *Vpreb3* (Pre-B lymphocyte gene 3, +3.2) and reduced *IL10* (Interleukin 10, −2.52), which has an anti-inflammatory capacity. Platelet-activating factor (1-alkyl-2-acetyl-sn-glycero-3-phosphocholine) is a potent immune response activator [100]. 

In a study by Díaz-Galván et al. [41], arachidonic acid metabolism was one of the main metabolic processes enriched with overexpressed genes due to BioCholine. This fatty acid and linoleic, eicosapentaenoic, or docosahexaenoic acid can be found in the sn2 position of phospholipids [101]. It is important to consider that the types of fatty acids provided in the diet can influence the types of fatty acids present in the phospholipids of blood cells [102], and BioCholine provides phospholipids in the form of phosphatidylcholine and presumably other conjugate choline compounds.

Phospholipids participate in essential cell signaling networks to maintain an effective innate immune response by recognizing molecules derived from their hydrolysis [101]. The contribution of phosphatidylcholine with BioCholine could have increased the expression of the *Pla2g2d*, *Pla2g4e*, and *Pla2g6* genes, which have cellular expression patterns and indicate greater activity of leukocyte cells and platelets (Figure 3). These genes code for type 2 phospholipases responsible for lysing phospholipids’ sn2 position, such as phosphatidylcholine, and releasing the polyunsaturated fatty acids contained therein [103]. Orzuna-Orzuna (unpublished data from the doctoral dissertation) and Roque-Jiménez et al. [20] measured the fatty acid profile of BioCholine and reported linoleic acid (C18:1 cis 9, 12) as the first long-chain fatty acid with concentrations of 58.94 and 65%, respectively.

Linoleic acid acts as a precursor of arachidonic acid [104], which increases the expression of the *Alox12* and *Alox15* genes that code for lipoxygenase enzymes expressed in eosinophils and platelets (Figure 3) [101]. These enzymes mainly use arachidonic acid (released by phospholipases) to generate monohydroperoxides (HpETEs), although they can also use docosahexaenoic or linoleic acid. This evidence indirectly shows the metabolite’s actions in the animal, suggesting the passing of some through the rumen by increasing the expression of genes in blood cells related to the hydrolysis of phosphatidylcholine. In addition to the genes that code for phospholipases, another example is the expression of the prostaglandin D2 synthase (*Ptgds*) and prostaglandin I2 (*Ptgis*) genes, which participate in pathways mediated by cyclooxygenases in immune cells from the oxidation of arachidonic acid (Figure 3) [101].

An increase in the flux of phosphatidylcholine into the blood increases the activity of innate immune cells. These identify the presence of phospholipids as an indicator of injuries (bleeding or trauma) activating homeostatic enzymatic mechanisms that will lead to the production of oxidized lipids that will start biological responses related to the inflammation process to prevent bacterial invasions and initiate the process of inflammation, wound healing, and repair [105].

However, the products derived from the action of phospholipases, lipoxygenases, monooxygenases, and cyclooxygenases act as agonists for activating the transcription factor of peroxisome proliferation alpha (*PPARα*). *PPARα* exhibits a potential anti-inflammatory capacity with a substantial impact on the physiology of the immune system by interfering with major inflammatory transcription factors and stimulating the catabolism of inflammatory precursors through fatty acid oxidation [106].

The increase in the expression of *PPARα* indicates a greater activity of innate immune cells, such as basophils, eosinophils, monocytes, and macrophages [107], but without leading to an inflammatory process. Thus, the flow of phosphatidylcholine into the blood with BioCholine in calves would strengthen the immune response [41] by increasing the efficiency of innate immune cells to recognize or census microorganisms [106].

### 3.11. Antioxidants

Variables that evaluate the polyherbal’s antioxidant properties have been registered in a few experiments. Alba et al. [35] reported improved antioxidant activity in milk (GPx 59.86% and GST 58.06%) over the control with Lacaune sheep supplemented with 5 g/d BioCholine. In addition, the serum GPx increased by 31.3% (*p* = 0.07) in samples collected between 7 and 45 days of lactation and reduced Thiobarbituric acid reactive substances assay (TBARS) by 6.25%, which are indicators of better antioxidant status. Alba et al. [36] confirmed the antioxidant effects (*p* < 0.01) in lactating Lacaune ewes at doses of 5 and 10 g/d. The authors observed a significant reduction in lipid peroxidation (LPO) and reactive oxygen species (ROS) in milk and serum sampled on days 15 and 20 with dietary BioCholine. 

The variables that indicate a greater antioxidant condition can be explained by the presence of several metabolites with antioxidant properties in BioCholine, such as 4-vinylguaiacol [19] and hexadecenoic acid methyl ester (C16:0) [20]. 4-vinylguaiacol has demonstrated an antioxidant effect [98], and C17:0 can protect DNA from oxidative damage induced by dietary fats [108].

Another hypothesis of antioxidant capacity would be synthesizing antioxidant molecules from metabolites with rumen escape capacity provided by the polyherbal, such as phosphatidylserine. Transcriptome analysis of hair lamb liver cells (Orzuna-Orzuna unpublished data from the doctoral dissertation) supplemented with BioCholine shows that serine could be used in the liver to synthesize cystathione by increasing the expression of the *Cbs* (+2.23) gene (which codes for cystathione β-synthetase). This effect would stimulate the expression of the *Gss* gene (+1.72) that codes for the glutathione synthetase enzyme responsible for synthesizing reduced glutathione via transsulfuration (Figure 4). This non-enzymatic molecule is one of the first lines of defense against oxidative damage [109].

In addition, BioCholine reinforces its antioxidant capacity by reducing the production of ROS by increasing the expression of the genes *Ndufc1*: 2.66, *Ndufa1*: 2.47, *Ndufb4*: 1.9, *Ndufv2*: 1.85, *Ndufa3*: 1.77 (complex I—NADH: ubiquinone oxidoreductase), *Uqcr10*: 1.99, *Uqcrfs1*: 1.87, *Uqcrq:* 1.78 (complex III—Cytochrome c reductase), and *Cox7a2l*: 1.55, *Cox6b1*: 1.51 (complex IV—Cytochrome c oxidase) (Figure 5). The increase in the expression of these genes involved in the electron transport chain would increase their oxidative efficiency in de-electronizing substrates, reducing the loss of electrons and diminishing their interaction with oxygen before it is reduced to water [110,111]. Accordingly, Cavaliere et al. [112] reported a close link between inflammation, redox status, and hepatic mitochondrial respiratory capacity, this last link due to an increased activity of carnitine palmitoyltransferase (CPT), the rate-limiting enzyme for fatty acid entry into the mitochondria.

These data indicate lower oxidative stress and agree with the genes downregulated by BioCholine: *Gsta3* (−2.86), *Gsta4* (−2.56), *Gstm3* (−2.51), and *Gstt4* (−1.57), together with *Aldh3a1* (−2.63) and *Adh4* (−2.15), which are related to lower glutathione conjugation with toxins in a xenobiotic detoxification process as part of drug metabolism—cytochrome P450 [113,114].

The ontological pathways modified with BioCholine as a cancer signaling network and the path of proteoglycans in cancer in the experiment by Díaz-Galván et al. [41], as well as in the pathway of chemical carcinogenesis in the investigation by Orzuna-Orzuna (unpublished data from doctoral dissertation), might be the result of better cellular functional integrity associated with antioxidant metabolites of the polyherbal. BioCholine also improved the response in cancer pathways in dogs compared to choline chloride [21].

### 3.12. Fertility

Little attention has been paid to changes in reproductive variables when using BioCholine. However, in addition to the genes referred to with energy metabolism, Díaz-Galvan et al. [41] showed the overexpression of some genes related to processes that affect reproduction, such as the PELP1 pathway—a novel estrogen receptor-interacting Protein (+4.0), the thyroid hormone signaling pathway X (+2.0), arachidonic acid metabolism (+2.9), and the ribosome pathway (+2.0) [41]. The multiyear evaluation with BioCholine showed that the feed plant additive improved fertility in cows during the first lactation (*p* < 0.01; 45.33% BioCholine vs. 37.0% control) [29]. 

This increase in the fertility observed in dairy cattle supplemented with BioCholine [29] could be related to the increase in the expression of genes that code for proteins responsible for synthesizing glutathione. Glutathione in males is an essential endogenous antioxidant responsible for the uptake of ROS in sperm and seminal plasma, protecting it from oxidative damage [115]. Zou et al. [116] added glutathione to the semen extender of Guanzhong dairy goats and observed improvements in sperm fertilization ability by reducing ROS levels.

In females, glutathione protects the ova from damage caused by oxidative stress during folliculogenesis; oocytes with higher levels of intracellular glutathione produce healthier and stronger embryos [117]. Adeoye et al. [118] pointed out that glutathione maintained the biological value of germ cells and implicated it in fertilization and the embryo’s early development.

In dairy cattle, high levels of oxidative stress can lead to dysregulation of reduced glutathione synthesis, manifesting in decreased milk production and reproductive disorders [119]. Given this scenario, the gene changes modulated by BioCholine can improve reproductive efficiency by reducing oxidative stress.

## 4. Conclusions

The results of experiments in which BioCholine was compared with RPC in small ruminants indicate that the polyherbal can be an alternative source to protected choline. The administration of BioCholine Powder in domestic ruminants’ diets improves productive performance variables, blood metabolite indicators of protein metabolism and liver health, and gene expression, which can be explained by all secondary metabolites, confirming its nutraceutical properties. The magnitude of the response depends on the dose and the species. The presence of all secondary metabolites in the polyherbal shows its impacts on energy protein metabolism in the animal methylation, health, and antioxidant status, as well as the net energy used from the diet.

## Figures and Tables

**Figure 1 animals-14-00667-f001:**
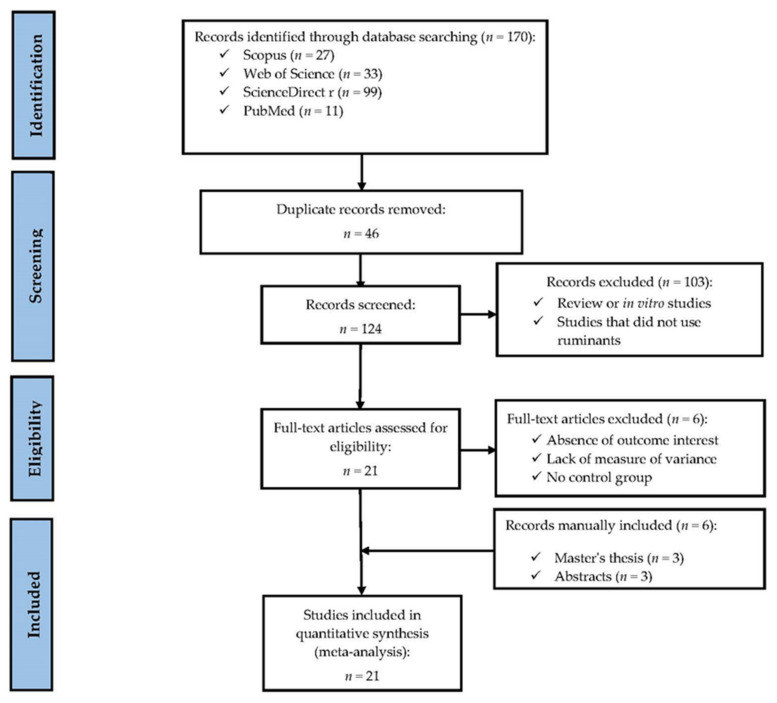
A PRISMA flow diagram detailing the literature search strategy and study selection for the meta-analysis.

**Figure 2 animals-14-00667-f002:**
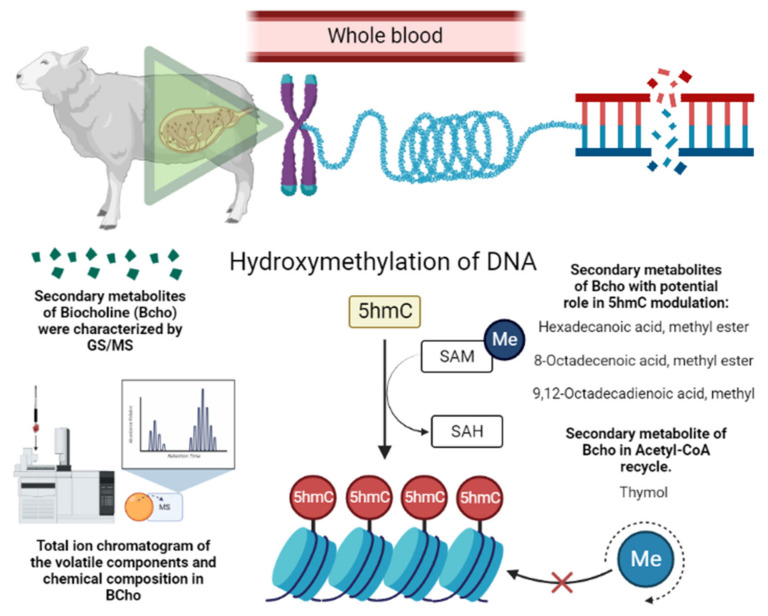
Presumable epigenetic mechanism of BioCholine over DNA methylation in ewes. SAM: S-Adenosyl methionine; SAH: S-Adenosyl homocysteine; Me: methionine; 5hmC: 5-Hydroxymethylcytosine.

**Figure 3 animals-14-00667-f003:**
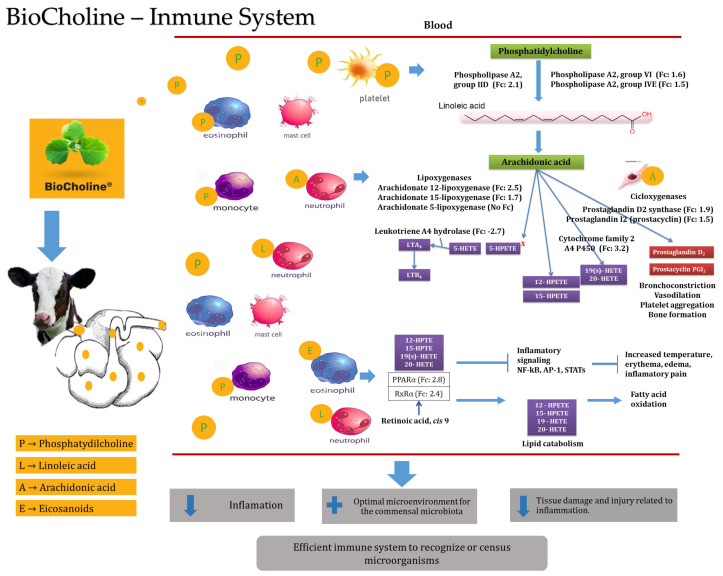
Presumable mode of action of BioCholine over innate immune response in weaning calves. LTA: leukotriene A4, LTB: leukotriene B4, *HpETE*: hydroperoxyeicosatetraenoic acid, *HETE*: hydroxyeicosatetraenoic acids, *PPARα*: peroxisome proliferator-activated receptors, *NF-kB*: nuclear factor kappa-light-chain-enhancer of activated B cells, AP-1: activator protein.

**Figure 4 animals-14-00667-f004:**
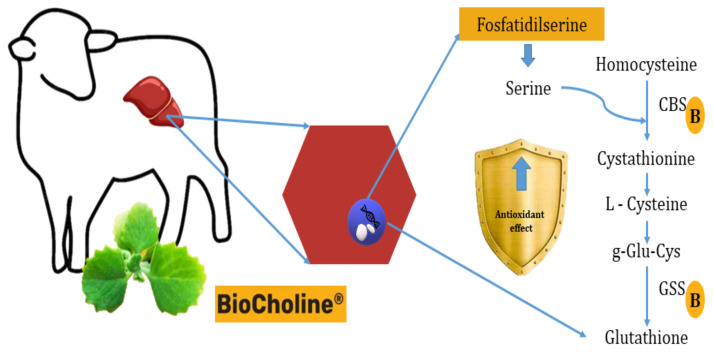
Presumable mode of action of BioCholine over antioxidant capacity in lamb hepatocytes by increasing the synthesis of reduced glutathione from serine and homocysteine. B: BioCholine^®^, CBS: cystathionine beta synthase, GSS: glutathione synthetase.

**Figure 5 animals-14-00667-f005:**
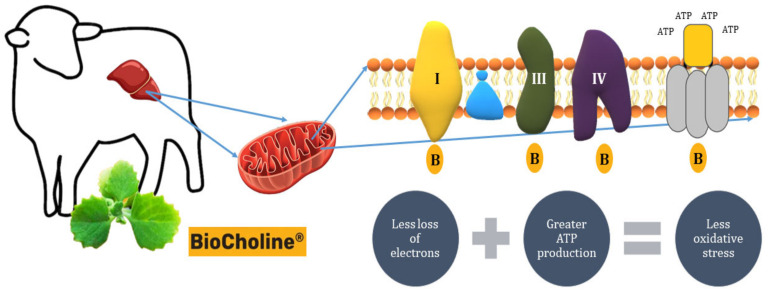
Presumable mode of action of BioCholine on the efficiency of the electron transport chain in hepatic mitochondria, increasing ATP synthesis and reducing the production of reactive oxygen species. B: BioCholine.

**Table 1 animals-14-00667-t001:** Description of the studies included in the meta-analysis database.

Reference	Duration, Days	Animal Specie	Dose (g/day)	Experimental Design
Alba et al. [35]	65	Ewes	0, 5	Completely randomized design (repeated measures)
Alba et al. [36]	20	Ewes	0, 5, 10	Completely randomized design (repeated measures)
Ayala-Monter et al. [37]	52	Ewes	0, 5, 10, 15	Completely randomized design (dose response)
Bárcena-Gama et al. [38]	45	Lambs	0, 6	Completely randomized design
Bello-Cabrera et al. [39]	60	Dairy Goats	0, 8	Completely randomized design
Cañada et al. [40]	90	Dairy Cow	0, 10, 20	Completely randomized design (dose response)
Crosby et al. [18]	60	Ewes	0, 4	Completely randomized design
Díaz-Galván et al. [41]	60	Calves	0, 3, 4, 5	Completely randomized design (dose response)
Estrada [42]	143	Ewes	0, 4, 8	Completely randomized design
Godinez-Cruz et al. [15]	42	Lambs	0, 4	Completely randomized design
Gutierrez et al. [29]	1095	Dairy Cows	0, 17	GLM linear mixed model (year-treatment fixed)
Leal et al. [43]	75	Lambs	0, 4, 8	Completely randomized design (repeated measures)
Martínez-Aispuro et al. [28]	52	Lambs	0, 5, 10, 15	Completely randomized design (dose response)
Martínez-García et al. [44]	42	Lambs	0, 4	Completely randomized design (repeated measures)
Mendoza et al. [19]	60	Dairy cows	0, 15	Completely randomized design
Morales et al. [45]	110	Dairy Goats	0, 4, 8	Completely randomized design (dose response)
Nunes et al. [46]	84	Dairy cows	0, 7, 14, 21	Latin Square (repeated measures)
Orzuna-Orzuna (Appendix A)	56	Lambs	0, 4, 7, 11	Completely randomized design (dose response)
Ortega-Alvarado et al. [47]	90	Dairy cows	0, 20, 40	Completely randomized design
Rodríguez-Guerrero et al. [48]	19	Ewes	0, 4	Completely randomized block design
Roque-Jimenez et al. [20]	35	Ewes	0, 4	Completely randomized design (repeated measures)
Suarez-Suarez et al. [49]	51	Ewes	0, 4	Completely randomized design

**Table 2 animals-14-00667-t002:** Growth performance and milk production of lambs and lactating ewes supplemented with BioCholine.

					Heterogeneity	
Item	N (NC)	Control Means (SD)	WMD (95% CI)	*p*-Value	*p*-Value	I^2^ (%)
DMI, kg/d	5 (9)	1.457 (0.376)	0.006 (−0.048; 0.060)	0.828	<0.001	88.87
ADG, kg/d	6 (11)	0.296 (0.089)	0.013 (0.003; 0.022)	0.012	0.255	19.73
FCR, kg/d	5 (9)	4.79 (0.454)	−0.068 (−0.246; 0.110)	0.454	<0.001	70.07
FBW, kg	6 (11)	45.03 (7.320)	1.250 (0.315; 2.185)	0.009	0.001	64.96
MP, kg/d	5 (10)	1.175 (0.617)	0.130 (0.000; 0.259)	0.049	<0.001	93.07

N: number of studies; NC: number of comparisons between the BioCholine treatment and control treatment; SD: standard deviation; WMD: weighted mean differences between control and treatments with BioCholine; CI: confidence interval of WMD; *p*-value to χ^2^ (Q) test of heterogeneity; I^2^: proportion of total variation of size effect estimates due to heterogeneity; DMI: dry matter intake; ADG: average daily gain; FCR: feed conversion ratio; FBW: final body weight; MP: milk production.

**Table 3 animals-14-00667-t003:** Subgroup analysis of the effect of BioCholine doses on lamb productive performance.

Item	NC	WMD (95% CI)	*p*-Value
DMI, subgroup dose			
≤5 g	5	0.002 (−0.053; 0.057)	0.948
>5 g	4	0.011 (−0.100; 0.123)	0.844
ADG, subgroup dose			
≤5 g	6	0.009 (−0.001; 0.020)	0.082
>5 g	5	0.020 (0.001; 0.040)	0.040
FCR, subgroup dose			
≤5 g	5	0.001 (−0.276; 0.278)	0.994
>5 g	4	−0.129 (−0.405; 0.147)	0.359
FBW, subgroup dose			
≤5 g	6	1.174 (−0.209; 2.557)	0.096
>5 g	5	1.258 (−0.013; 2.529)	0.052

NC: number of comparisons between the BioCholine treatment and control treatment; CI: confidence interval of WMD; *p*-value to χ^2^ (Q) test of heterogeneity; DMI: dry matter intake; ADG: average daily gain; FCR: feed conversion ratio; FBW: final body weight.

**Table 4 animals-14-00667-t004:** Subgroup analysis of the effect of doses and starting times of BioCholine supplementation on sheep milk production.

Item	NC	WMD (95% CI)	*p*-Value
Subgroup dose			
4 (or 5 g/d)	7	0.148 (−0.036; 0.333)	0.115
8 (10 g/d)	3	0.077 (−0.132; 0.286)	0.471
Subgroup			
Prepartum	5	0.128 (−0.015; 0.270)	0.079
Postpartum	5	0.140 (−0.232; 0.512)	0.461

NC: number of comparisons between BioCholine treatment and control treatment. CI: confidence interval of WMD; *p*-value to χ^2^ (Q) test of heterogeneity.

**Table 5 animals-14-00667-t005:** Blood metabolites of ruminants supplemented with BioCholine.

					Heterogeneity	
Item	N (NC)	Control Means (SD)	WMD (95% CI)	*p*-Value	*p*-Value	I^2^ (%)
Glucose, mg/dL	12 (23)	67.47 (19.65)	0.466 (−4.312; 5.243)	0.848	<0.001	93.59
Cholesterol, mg/dL	10 (18)	91.75 (40.97)	2.941 (−3.105; 8.987)	0.340	<0.001	89.01
Triglycerides, mg/dL	7 (13)	55.10 (36.80)	−0.059 (−2.499; 2.381)	0.962	<0.001	77.66
β-Hydroxybutyrate, mmol/L	3 (7)	0.416 (0.27)	−0.000 (−0.044; 0.043)	0.982	0.001	72.98
Urea, mg/dL	6 (12)	44.18 (11.23)	−3.957 (−6.971; −0.944)	0.010	<0.001	81.61
Total protein, g/dL	11 (21)	6.91 (1.03)	0.167 (−0.041; 0.375)	0.115	<0.001	58.38
Albumin, g/dL	9 (18)	3.24 (0.74)	0.187 (0.052; 0.321)	0.007	<0.001	67.21
Globulin, g/dL	9 (19)	3.62 (0.57)	0.075 (−0.149; 0.300)	0.512	<0.001	84.88
AST, UI/L	7 (13)	73.20 (43.70)	−3.675 (−8.582; 1.232)	0.142	<0.001	87.76
ALT, UI/L	3 (5)	16.20 (1.45)	0.724 (−0.136; 1.583)	0.099	0.564	0.00

N: number of studies; NC: number of comparisons between the BioCholine treatment and control treatment; SD: standard deviation; WMD: weighted mean differences between control and treatments with BioCholine; CI: confidence interval of WMD; AST: aspartate transferase; ALT: alanine transaminase; *p*-value to χ^2^ (Q) test of heterogeneity; I^2^: proportion of total variation of effect size estimates that is due to heterogeneity.

**Table 6 animals-14-00667-t006:** Subgroup analyses on the effect of daily doses of BioCholine on serum metabolites of ruminants.

Item	NC	WMD (95% CI)	*p*-Value
Glucose, mg/dL			
≤4 g	10	−4.149 (−9.441; 1.144)	0.124
5–8 g	9	−1.163 (−9.657; 7.331)	0.788
>8 g	4	16.054 (−2.235; 34.344)	0.085
Cholesterol, mg/dL			
≤4 g	7	0.857 (−9.864; 11.579)	0.875
5–8 g	7	−0.414 (−7.198; 6.370)	0.905
>8 g	4	14.067 (2.898; 25.237)	0.014
Triglycerides, mg/dL			
≤4 g	4	0.785 (−5.549; 7.118)	0.808
5–8 g	5	0.179 (−3.681; 4.039)	0.928
>8 g	4	−1.333 (−3.437; 0.770)	0.214
β-Hydroxybutyrate, mmol/L			
≤4 g	4	−0.030 (−0.079; 0.019)	0.227
5–8 g	3	0.036 (−0.014; 0.087)	0.158
>8 g	-	-	-
Urea, mg/dL			
≤4 g	5	−6.656 (−10.550; −2.763)	<0.001
5–8 g	6	−2.124 (−7.594; 3.346)	0.447
>8 g	-	-	-
Total protein, g/dL			
≤4 g	9	0.126 (−0.151; 0.403)	0.371
5–8 g	8	−0.014 (−0.204; 0.177)	0.886
>8 g	4	1.042 (0.009; 2.074)	0.048
Albumin, g/dL			
≤4 g	7	0.264 (−0.013; 0.542)	0.062
5–8 g	7	0.028 (−0.099; 0.155)	0.667
>8 g	4	0.309 (−0.007; 0.625)	0.055
Globulin, g/dL			
≤4 g	7	−0.031 (−0.476; 0.415)	0.892
5–8 g	8	−0.085 (−0.373; 0.204)	0.562
>8 g	4	0.739 (0.122; 1.355)	0.019
AST, UI/L			
≤4 g	6	−5.682 (−14.417; 3.052)	0.202
5–8 g	6	−2.103 (−6.347; 2.142)	0.332
>8 g	-	-	-
ALT, UI/L			
≤4 g	2	0.296 (−2.206; 2.798)	0.817
5–8 g	2	0.666 (−0.980; 2.311)	0.428
>8 g	-	-	-

NC: number of comparisons between the BioCholine treatment and control treatment; CI: confidence interval of WMD; AST: aspartate transferase; ALT: alanine transaminase; *p*-value to χ^2^ (Q) test of heterogeneity.

**Table 7 animals-14-00667-t007:** Subgroup analysis of the effect of animal species on serum metabolites of ruminants supplemented with BioCholine.

Item	NC	WMD (95% CI)	*p*-Value
Glucose, mg/dL			
Goats	2	−7.985 (−10.758; −5.212)	<0.001
Calves	3	−19.373 (−28.177; −10.570)	<0.001
Lambs	11	9.797 (1.423; 18.171)	0.022
Sheep	7	−2.596 (−5.934; 0.743)	0.128
Cholesterol, mg/dL			
Goats	2	−23.715 (−36.697; −10.733)	<0.001
Calves	-	-	-
Lambs	9	11.537 (1.179; 21.895)	0.029
Sheep	7	−0.869 (−5.247; 3.510)	0.697
Triglycerides, mg/dL			
Goats	-	-	-
Calves	-	-	-
Lambs	8	0.780 (−2.647; 4.207)	0.656
Sheep	5	−1.663 (−3.653; 0.327)	0.102
β-Hydroxybutyrate, mmol/L			
Goats	2	0.195 (−0.070; 0.460)	0.149
Calves	3	−0.027 (−0.087; 0.032)	0.367
Lambs	-	-	-
Sheep	2	0.022 (−0.011; 0.055)	0.185
Urea, mg/dL			
Goats	2	−3.740 (−12.474; 4.994)	0.401
Calves	3	−9.410 (−10.782; −8.038)	<0.001
Lambs	2	1.605 (−2.805; 6.015)	0.476
Sheep	5	−0.100 (−2.890; 2.690)	0.944
Total protein,			
Goats	2	−0.115 (−0.478; 0.248)	0.534
Calves	3	0.010 (−0.237; 0.257)	0.937
Lambs	11	0.437 (0.034; 0.840)	0.034
Sheep	5	0.071 (−0.186; 0.329)	0.587
Albumin, g/dL			
Goats	2	0.305 (−0.332; 0.942)	0.348
Calves	3	0.207 (0.020; 0.393)	0.030
Lambs	8	0.243 (0.004; 0.482)	0.046
Sheep	5	0.027 (−0.143; 0.197)	0.758
Globulin, g/dL			
Goats	2	−0.095 (−0.385; 0.195)	0.521
Calves	3	−0.233 (−0.370; −0.097)	<0.001
Lambs	9	0.454 (0.073; 0.836)	0.020
Sheep	5	−0.115 (−0.511; 0.281)	0.570
AST, UI/L			
Goats	2	−0.355 (−10.851; 10.141)	0.947
Calves	3	−4.457 (−6.517; −2.396)	<0.001
Lambs	3	−12.436 (−50.482; 25.610)	0.522
Sheep	3	3.667 (−0.198; 7.532)	0.063
ALT, UI/L			
Goats	-	-	-
Calves	-	-	-
Lambs	2	0.417 (−1.137; 1.972)	0.599
Sheep	2	0.725 (−0.647; 2.097)	0.300

NC: number of comparisons between the BioCholine treatment and control treatment; CI: confidence interval of WMD.

**Table 8 animals-14-00667-t008:** Net energy values for treatments estimated from lamb performance in experiments with RPC and BioCholine (4 g/d).

	Control	BioCholine	RPC	Author	
NEm Mcal/kg	1.833	1.878	1.757	Martínez-García et al. [44]	
NEg Mcal/kg	1.198	1.237	1.131		
NEm Mcal/kg	1.697	1.817	1.811	Godínez-Cruz et al. [15]	
NEg Mcal/kg	1.078	1.184	1.17		
Weighted means	Control	BioCholine	Effect Size %	Chi-squared	*p*-value
NEm Mcal/kg	1.765	1.848	+4.67	0.741	0.3893
NEg Mcal/kg	0.645	0.655	+1.62	0.253	0.6149
Σ n	16	16			
Weighted means	Control	RPC	Effect Size %	Chi-squared	*p*-value
NEm Mcal/kg	1.765	1.784	+1.08	0.168	0.6816
NEg Mcal/kg	0.645	0.645	+0.02	0.003	0.9555
Σ n	16	16			

**Table 9 animals-14-00667-t009:** Net energy values for treatments estimated from lamb performance in experiments with grade levels of BioCholine and effect size observed with the polyherbal.

	BioCholine Intake g/d					
	0	4 (5)	8 (10)	15	Total N (Sex)	Author
NEm Mcal/kg	1.99	2.029	2.063	2.215	40 (males)	Martínez-Aispuro et al. [28]
NEg Mcal/kg	1.335	1.369	1.399	1.533		
NEm Mcal/kg	1.414	1.533	1.569		48 (females)	Leal et al. [43]
NEg Mcal/kg	0.83	0.935	0.966			
NEm Mcal/kg	1.711	1.73	1.778	1.647	52 (females)	Ayala-Monter et al. [37]
NEg Mcal/kg	0.757	0.764	0.809	0.716		
Weighted means	Control	BioCholine	Effect size %	Chi-squared	*p*-value	
NEm Mcal/kg	1.661	1.779	+7.15	2.915	0.0878	
NEg Mcal/kg	0.935	1.022	+9.25	3.833	0.0502	
Σ n	39	101				

**Table 10 animals-14-00667-t010:** LS means of live weight changes by BioCholine supplementation in ewes supplemented with 4 g/d from the reanalyzed data of Roque-Jiménez et al. [20].

	Control	Third	All Gestation	SEM	Contrast *p*-Value	
					I	II
Lactating Ewe						
Ewe birth weight (kg)	64.00	61.33	61.66	3.082	0.92	0.93
BW on week 5 lactation (kg)	52.66	51.00	53.33	2.368	0.85	0.13
Live weight changes (kg/d)	−0.323	−0.293	−0.240	0.0295	0.92	0.02
Milk yield (kg/d)	907	713	1142	53.06	0.0004	0.0021
Offspring						
Birth lamb weight (kg)	4.63	6.45	5.67	0.4626	0.04	0.15
Weaning lamb weight (kg)	9.14	12.35	13.55	0.6856	0.68	0.60
Daily weight changes (kg/d)	0.196	0.166	0.223	0.0171	0.08	0.59

I. Control vs. supplemented; II. last third vs. all gestation. Due to the nature of the experiment with fetal blood sampling, the number of replications for the presented variables is small (n = 3).

**Table 11 animals-14-00667-t011:** Results reported with BioCholine in dairy cattle and effect size of the polyherbal.

	BioCholine Intake g/d					
	0	7–10	14–15	17–21	40	Author
Milk (kg/d)	18.98	19.67		20.06		Cañada et al. [40]
Milk kg/d	40.01			36.73	40.16	Ortega-Alvarado et al. [47]
Milk (kg/d)	35.81			36.37		Gutiérrez et al. [29]
Milk (kg/d)	29.5	30.5	30	29.8		Nunes et al. [46]
Weighted means	Control	BioCholine	Effect Size %	Chi-squared	*p*-value	
Milk Production	35.50	35.72	+0.59	7.884	0.005	
Σ n	1436	1325				

## Data Availability

The datasets used and analyzed in the current study are available.

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
