# Peer review of "A Polyherbal Mixture with Nutraceutical Properties for Ruminants: A Meta-Analysis and Review of BioCholine Powder"

_animals, 2024, doi:10.3390/ani14050667_

Round 1

Reviewer 1 Report

Comments and Suggestions for Authors

Comments and suggestions are reported in the attached file 

Author Response

Please see the attacment

Reviewer 2 Report

Comments and Suggestions for Authors

Manuscript number: Animals 2831771

Title: A Polyherbal Mixture with Bypass Phosphatidylcholine for Ruminants: Meta-Analysis and Review of Biocholine Powder

General comment:

            Despite its commendable writing and grammar, this journal still contains numerous ambiguities. The initial explanation pertaining to the terms biocholine, RPC, and Polyherbal Mixture is not sufficient, potentially causing cognitive delays in the reader. Additionally, inconsistent word usage compels readers to reconsider the definitions of the terms. The stated objective fails to answer the question posed in the title, and similarly, the conclusion fails to address the stated objective. The journals selected according to the PRISMA method for meta-analysis are opaque and not listed in the table, preventing anyone from carrying out the validation process of the meta-analysis results. There were 23 papers in the database that missed a description of the study title and information. The research that is to be published in the "Animals Journal" should possess the qualities of clarity and interest.

Abstract:

- See comment in a section of the text.

- Keyword: Plant feed additives? Should be similar to title!

Introduction:

· L 60-61: A brief description of rumen-protected choline (RPC) is necessary.

· L 66-69: This phrase indicates that RPC is administered to dairy cows, but it does not include information on dairy cows specifically. Instead, it discusses the beneficial outcomes seen in dairy goats, beef cattle, and feedlot sheep. What is the status of dairy cows? To clarify the influence on dairy cattle, it is advisable to merge this statement with line 61.

· L 107-109: Provide a reference.

· L 107-109: Provide literary references that explore this particular parameter.

· L 109-114: This objective should be succinctly condensed into a single impactful line and does not need redundant explanations since it may introduce prejudice.

· L 112-115: The objective focuses on elucidating the impacts of polyherbal supplementation by meta-analysis. However, this is inconsistent with the title, which specifically pertains to a polyherbal mixture including Bypass Phosphatidylcholine. Make it simple.

· General: Meta-analyses are conducted to draw conclusions by synthesizing the findings of multiple studies; however, these conclusions are, themselves, derived from the results of numerous inconsistent studies. Prior investigations concerning Polyherbal Mixture with Bypass Phosphatidylcholine appear to be contradictory. Kindly specify the urgency.4

· It is important to bring out this sense of urgency prior to discussing objectives.

Material and method:

L 119-125: According to the PRISMA method, what is the maximum number of journals that can be accepted from each journal platform? What number of journals lack relevance? It is essential for the PRIMA method exhibit clarity when selecting the database!

L 123-124: There are 2 unpublished data documents from our research team included as a database for meta-analysis, how can this data be considered valid if the references have not been published? Please explain more clearly.

L 127-128: On the basis of the PRISMA method, the selection process and acceptable inclusion criteria should be specified.

L 140-141: WMDs or WMD, please consistent!

L 150-153: The text is complicated, including many conjunctions (and) and commas (,).

L 155-159: There is no need for a clear explanation of the terms "control" and "treatment." For instance, the control group does not use biocholine, whereas the treatment group incorporates the use of biocholine. It should be clear!

Result and Discussion

L 193-194: Provide a reference

L 204-205: Adjust the margins

L 220-242: Exemplifying an excessive number of research findings with only one citation; further citations are required to bolster the argument.

L 254-257: The paragraph is insufficient in length to provide a comprehensive explanation.

L 312-314: By setting fixed margins, pictures may be repositioned while ensuring that excessive white space is avoided.

L 341-344: Non-public data lacks sufficient strength due to the inability to verify its accuracy.

L 349-420: Provide information and a list of accepted journals so that you can continue in the meta-analysis process. Because “References are used for constructing the meta-database” it is really important to know. Data is difficult to trust if there is no transparency about the database.

L 431-432: Provide a reference about this

L 451: How does a minimal quantity of repetitions impact the final score?

L 453: What species of animal is the subject of discussion? Write with clarity.

L 458-461: Is the value significant? (P<0.10)?

L 467-478: It appears to merely provide additional details and fails to elucidate the findings presented in Table 8.

L 491-492: The explanation for the data in Table 9 is absent from Table 9. Please elaborate.

L 508-510: The animal's genetic response can be an explainable problem in milk production.

L 629-633: Provide the reference

L 639: TBARS meaning?

Conclusion:

The conclusion fails to address the stated objectives. While milk production is alluded to in the introduction, it is not elaborated upon.

References:

Check format style and Journal Abbreviation!

Comments on the Quality of English Language

Moderate editing of English language required

Reviewer 3 Report

Comments and Suggestions for Authors

The aim of the manuscript was to review the state of art/science on the application of BIOcholine- a polyherbal product containing Choline as a dietary additive in ruminants. There is evidence that rumen protected choline have beneficial effects on ruminants although there is denovo synthesis by rumen microbes.

The current manuscript used a narrative review as well as meta-analysis to analyse effect size from available experiments published and unpublished.

In the introduction section, the review should include discussion on the de novo synthesis of choline in ruminants and establish the net choline supply to justify supplementation? Will exogenous supplementation be effective in all cases?

The manuscripts included in the meta-analysis, although referenced, but are not included as a list/table. Provide a descriptive summary table of the manuscripts included in the meta-analysis showing the paper authors, main parameters, experimental design etc. What were the controls in these studies? Which animal subjects were used and what experimental design was used? What was the duration of feeding before data on choline effect was measured? Without these key factors, the authors will have embarked on a skewed narrative around the BIoCHoline efficacy. Was the control no- biocholine or a synthetic choline in the studies reported?RPC was not the control or compared alternative in all studies.

Equally, the use of keywords are not fully descriptive. The authors listed the keywords but not how they were combined. There is no guarantee that another research team will combine these keywords in a different set of ways and get the same output in terms of Journal included in the review.

I have not come across the inclusion of unpublished data and student projects/dissertations in meta-analysis. This attempt by the current authors appears to be a biased attempt to project an outcome. Me-analysis by its nature intend to rely on evidenced peer review outcomes to summarise the state of knowledge on a subject area- rumen bypass choline using ethnobotanicals. This may be used in the narrative review section. Meta-analysis seems to have a more stricter conde of conduct but the authors may have violated some of them.

Other comments

L333: difference- is this a decrease or increase in these VFA component?

L454-459: This looks more like a systematic review of the effect of doses rather than a meta-analysis. The author may not/ may not have been able to pull the dose across papers together and regress but validate this hypothesis based on reference [27].

L480-484: Were the controls no choline or synthetic choline?

L526-527: Health indicators improved. The authors of the study did not estimate treatment cost did they? At best, the review can speculate potential improvement in health status, antioxidant profile etc. That may not be equivalent to to reduced treatment cost.

L550: differences between what and what? Across individual animals? Associated to what? This sentence should be rephrased.

L729: polyherbal has metabolites that can be bypassed, impacting energy–protein metabolism (methylation), health, and antioxidant status, as well as the net energy used from the diet”. The sentence is not clear, especially the word bypassed. “Polyherbal has metabolites that bypassed rumen degradation, thus impacting energy-protein metabolism in the animal-methylation, health…”. Could this be an appropriate correction?

 Overall, the manuscript should be revised, applying more stringently, the principles of carrying  out a meta-analysis. The manuscript has good merit and would enjoy interest of readers.

Comments on the Quality of English Language

There are only minor language issues. The manuscript was well-written.

Round 2

Reviewer 2 Report

Comments and Suggestions for Authors

My comments were addressed well and no further suggestions! Congratulations!

Author Response

We appreciate the observations and responded to the suggestions.

Reviewer 3 Report

Comments and Suggestions for Authors

The authors have provided a detailed revision of the manuscript. One major issue of concern is the inclusion criteria as detailed below.

Line 64: define RPC at first use.

 Line 131-132: Abstracts do not provide full experimental details. I wonder how these made the inclusion criteria. Equally, student dissertations are not publicly available. I am not sure if these agrees with the PRISMA methodology referenced in Line 128. Figure 1 shows full text articles. No abstracts of papers included.

One of the key criteria for a meta-analysis is that the search criteria is transparent and the experimental procedure is repeatable. I have expressed reservations about using studies/data obtained from dissertation databases of universities in carrying out a meta-analysis. Can this be repeatable? The authors have not provided any validated example where this is done. This makes repeatability of the current study doubtful. This is a strong point that must not be glossed over. I think authors may limit those sources to the narrative review components and strictly use only publicly available sources for the meta-analysis.

L225: correct spelling error.

Comments on the Quality of English Language

The paper is grammatically ok.

Author Response

(The authors gave the same response as above.)
